# Bacteria use structural imperfect mimicry to hijack the host interactome

**Natalia Sanchez de Groot[1]\*, Marc Torrent Burgas[2]\***

**1** Gene Function and Evolution Lab, Centre for Genomic Regulation (CRG), Dr. Aiguader 88, Barcelona, Spain, **2** Systems Biology of Infection Lab, Department of Biochemistry and Molecular Biology, Biosciences Faculty, Universitat Autònoma de Barcelona, Cerdanyola del Vallès, Spain

\* natalia.sanchez@crg.eu (NSdG); marc.torrent@uab.cat (MTB)

## Abstract

Bacteria use protein-protein interactions to infect their hosts and hijack fundamental pathways, which ensures their survival and proliferation. Hence, the infectious capacity of the pathogen is closely related to its ability to interact with host proteins. Here, we show that hubs in the host-pathogen interactome are isolated in the pathogen network by adapting the geometry of the interacting interfaces. An imperfect mimicry of the eukaryotic interfaces allows pathogen proteins to actively bind to the host's target while preventing deleterious effects on the pathogen interactome. Understanding how bacteria recognize eukaryotic proteins may pave the way for the rational design of new antibiotic molecules.

**Data Availability Statement:** All relevant data are within the manuscript and its Supporting Information files.

**Funding:** Ministerio de Economía y Competitividad (MEC):Marc Torrent RYC-2012-09999; Ministerio

## Author summary

To fulfil their function, proteins need to interact with each other forming complexes. Understanding how pathogen proteins bind their host counterparts is important to explain how bacteria can infect, survive and proliferate inside cells. To achieve that, pathogen proteins mimic eukaryote interfaces to interact with the host. However, we discovered that such mimicry is imperfect, and pathogen proteins display particular features that are not found in eukaryotic complexes. This imperfect mimicry would allow pathogen proteins to actively bind to the host targets while preventing deleterious effects on the pathogen network. Indeed, we show that highly connected proteins (hubs) in the host-pathogen networks are mostly isolated in the pathogen network. The existence of imperfect mimicry opens the door to the design of new molecules aimed to target host-pathogen complexes with reduced side effects. Hence, in the long term, our results may lay the foundation of a new class of antimicrobials.

## Introduction

In nature, bacteria do not exist in isolation but within communities of multiple species that require the bacteria to communicate and organize [1]. In particular, pathogenic bacteria need to interact with host cells to ensure their survival and proliferation [2]. Most of these interactions are mediated by protein effectors that are involved in microbial virulence [3, 4]. These

de Economía y Competitividad (MEC):Marc Torrent SAF2015-72518-EXP; Ministerio de Economía y Competitividad (MEC):Marc Torrent SAF2017-82158-R; European Society of Clinical Microbiology and Infectious Diseases (ESCMID): Marc Torrent ESCMID-2016 The funders had no role in study design, data collection and analysis, decision to publish, or preparation of the manuscript.

**Competing interests:** The authors have declared that no competing interests exist.

effectors, delivered to the host by secretion systems [5], outer membrane vesicles [6] or other specific mechanisms [7], allow bacteria to bind to host cells, replicate and spread within the host and subvert its immune system.

The collection of all these interactions between the host and the pathogen are known as the host-pathogen interactome. In the interactome, proteins with high connectivity degree are known as interactome "hubs" and are associated with essential and conditional phenotypes [8, 9]. The correlation between node degree and gene essentiality is known as the centrality-lethality rule [10] and has been observed in many organisms and interspecies interactomes [11]. In infectious diseases, pathogens selectively target hubs in the host interactome to rewire specific pathways for their own benefit [8]. The elucidation of these interactions is of utmost importance to understand how pathogens hijack the host systems.

Protein interactions are mostly mediated by domain-domain pairs. However, the number of protein folds [12] and interface geometries [13] is limited and these structures are frequently 'reused', allowing a single protein to bind many partners. This recycling strategy is necessary because the interaction space is limited [14] and the distribution of domains in bacteria and eukaryotes has major differences. For example, the most common folds in prokaryotes are those involved in housekeeping functions, such as P-loop-containing NTPases and TIM barrels, whereas the distribution in eukaryotes is dominated by domains with regulatory functions, such as protein kinases and β-propellers [15]. Hence, these constraints set a diversity upper limit and impose restrictions on how bacteria can bind eukaryote proteins.

Despite recent advances in the characterization of host-pathogen interactions, the structural knowledge of these protein complexes is very limited and largely restricts our ability to understand pathological systems. Here, we analyzed the degree centrality of bacterial proteins in the pathogen and the host-pathogen interactomes and investigate the structural characteristics of the interactions involved. We observed that hubs in the host-pathogen interactome are largely isolated in the pathogen interactome. This behavior can be explained by an imperfect mimicry of host interfaces that allow bacteria to minimize the toxicity of these proteins by restricting the number of interactions while maintaining an affinity for host proteins.

## Results

### Hubs in the host-pathogen interactome are segregated in the pathogen interactome

Because effectors target specific host interactions, their interfaces need to be fine-tuned to interact with a precise set of targets [16]. If proteins are not controlled for promiscuous interactions, this can lead to toxic effects [17]. In the case of effectors, optimized interfaces must be controlled to avoid undesired pathogen self-interactions that may compromise cell performance. A safe ward strategy against unwanted interactions is the timely expression of proteins under certain circumstances [18]. If bacterial effectors were only produced when bacteria are in close contact with the host, their deleterious effects could be minimized. However, this does not seem to be the case. The expression of effectors delivered by secretion systems is not triggered upon infection in cases where data is available (**S1A Fig**). Also, other mechanisms of delivering virulence factors, such as extracellular vesicles, encapsulate large amounts of proteins from the bacteria cytoplasm and periplasm and do not specifically select their cargo [19]. This strategy would also be non-optimal as detection and killing of bacteria by the innate immune cells is very fast [20] while altering the expression of a protein may take longer [21].

In this context, either these proteins are physically isolated from the rest of the proteome or are integrated into the network in a controlled manner. In eukaryotes, spurious interactions are prevented by segregating proteins into different compartments. However, bacteria lack

these compartments. A possible alternative would be the use of protein condensates [22], non-membrane bound structures formed by liquid-liquid phase separation [23]. These condensates would allow proteins to be relatively isolated from the cell milieu, but ready to be delivered when required. However, we did not find a clear difference in condensate propensity between secreted effectors and the rest of the proteome (**S1B Fig**). Besides, bacteria are more proficient in forming solid aggregates, that are less dynamic than liquid condensates [24, 25].

Based on these evidences, we hypothesized that effectors could be integrated into the bacteria network in such a way that they were not deleterious when pathogens replicate outside the host but can be effectively deployed upon infection. By isolating effectors from the pathogen network, they would have less control over the interactome, minimizing the side effects. In agreement with this reasoning, we observed that effector proteins are significantly depleted of pathogen hubs (**Fig 1A**). These results were also validated in an orthogonal database to control for potential methodological bias (**Fig 1A**, see Materials and Methods for a detailed description of the control database).

To further investigate the integration of effectors in protein networks we compared its degree centrality both in the pathogen and the host-pathogen interactomes (**Fig 1B**). We were able to separate proteins into three clusters: (*i*) proteins that have a high number of interactions in the host-pathogen interactome but are highly isolated in the pathogen interactome (C1 cluster, **Fig 1C**), (*ii*) proteins that are isolated in the host-pathogen interactome but largely connected in the pathogen interactome (C2 cluster, **Fig 1C**) and (*iii*) proteins that are mainly isolated in both interactomes (C3 cluster, **Fig 1C**).

We investigated whether the structural properties of these proteins may explain the differences observed between clusters (**Fig 1D**). We found that the proteins belonging to the C1 cluster are richer in coil structure compared to proteins in clusters C2 and C3. This is consistent with an increased propensity to disordered regions, a property commonly observed in eukaryotic proteins. The proteins that define the C2 cluster tend to form alpha helix, which favors the binding to nucleic acids. Finally, the proteins belonging to the C3 cluster are enriched in beta-sheet structure and more prone to aggregate. Overall, these clusters show specific structural properties and may reflect the differences in the cellular milieu between prokaryotes and eukaryotes.

The fact that we did not find any cluster with a high number of interactions in both interactomes suggests that protein interfaces in the host and the pathogen are nearly orthogonal, meaning that is difficult to optimize an interface to act as a hub in the pathogen and the host-pathogen interactomes at the same time. These observations raise a fundamental question: how can bacteria discriminate between self and non-self interfaces?

## Bacteria use structural imperfect mimicry to interact with the host

Protein interactions are mediated by non-covalent bonds between residues located in the interaction core, which is a central area excluded from the solvent. This core region is surrounded by the rim area, which is partially buried, helps to exclude water molecules from the core and is involved in the modulation of the interaction. The core explains most of the binding energy of the complex, while the rim can tune the binding strength [26], particularly in small complexes [27].

To understand how the interface structure can be used to discriminate self and non-self interfaces in bacteria, we mined the PDB for bacteria-eukaryote (BE) complexes and found 89 nonredundant entries (**Fig 2A**, **S1 Table**). The BE dataset used is enriched in pathogenic proteins such as effectors, toxins and adhesins (**S1 Table**). Bacterial proteins included in the BE set are significantly upregulated in infection and are relevant for pathogenesis (**S2 Fig**). We

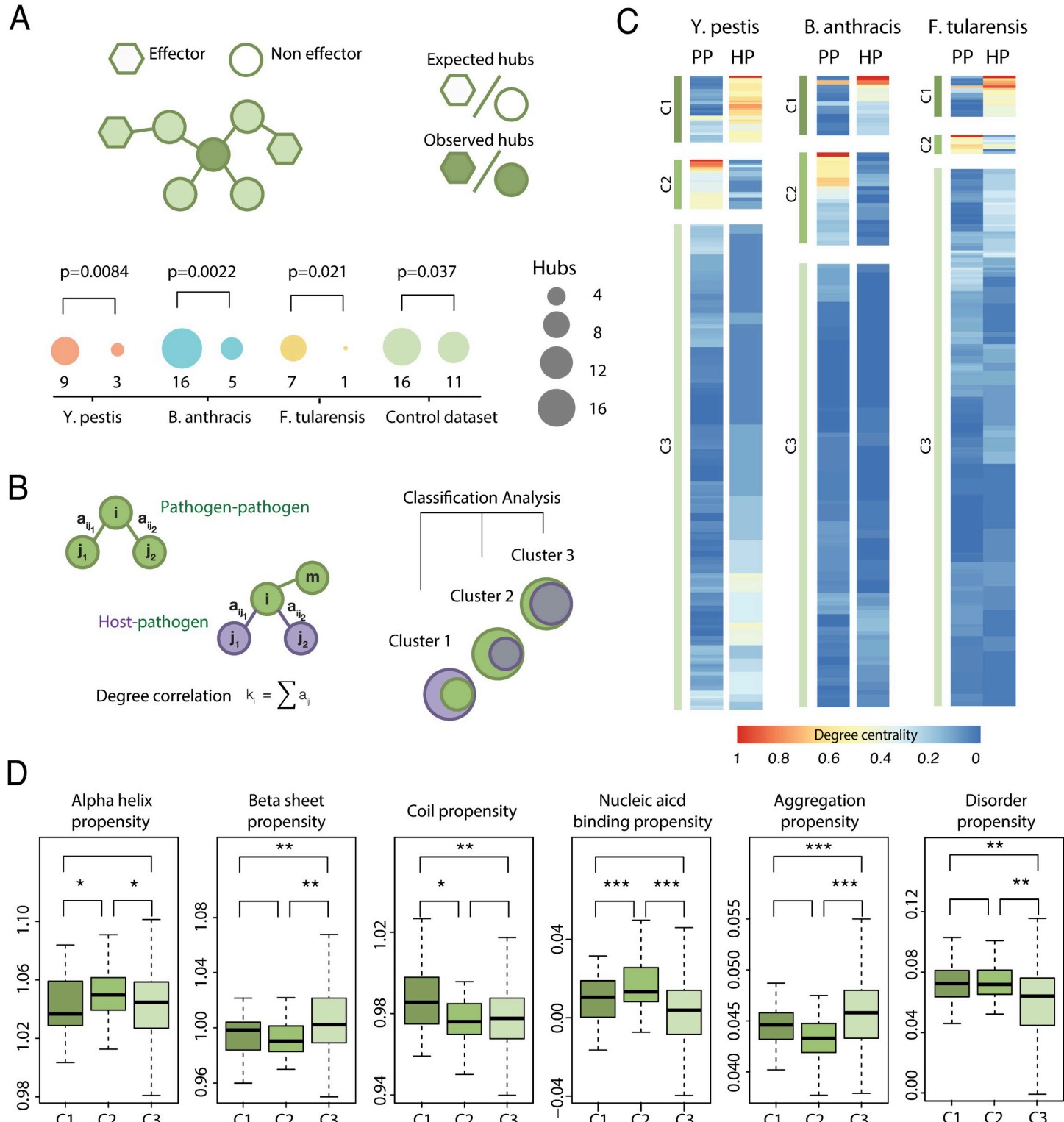

**Fig 1. Hubs in the host-pathogen interactome are largely isolated in the pathogen network.** (A) We computed the degree of interaction for all proteins in *Y. pestis*, *B. anthracis* and *F. tularensis* and asked whether the number of hubs (here defined as the 5% of most connected proteins) differ between effectors and non-effectors. In all cases, the number of hubs observed is significantly lower than expected. Similar results were obtained in the control database. The p-values were computed using a $\chi^2$-square test of independence to assess the probability of observing such a large discrepancy (or larger) between observed and expected values. Effectors were defined using EffectiveDB with a stringency threshold of 0.95 [52]. All comparisons remain statistically significant in the threshold interval 0.90–0.99 (p<0.05). (B) We compared the degree centrality of bacterial proteins in the pathogen and the host-pathogen interactomes. Based on the results obtained, we classified the bacterial proteins in clusters,

according a k-means clustering algorithm. (C) Based on this clustering, three different groups were identified: proteins that have a high number of interactions in the host-pathogen interactome but are highly isolated in the pathogen interactome (C1 cluster), proteins that are isolated in the host-pathogen interactome but deeply connected in the pathogen interactome (C2 cluster) and proteins that are mainly isolated in both clusters (C3 cluster). (D) The three clusters identified previously have distinctive structural properties. Proteins in C1 cluster are enriched in coil structure, which favors the presence of disordered regions; proteins in C2 cluster are enriched in alpha helix structure, which favors interaction with nucleic acids and proteins in C3 cluster are enriched in beta sheet structure, that favors aggregation. * p<0.05; ** p<0.01; ***p<0.005 using a Mann–Whitney U-test with α = 0.05.

also selected 183 bacteria-bacteria (BB) and 686 eukaryote-eukaryote (EE) complexes for comparison (**S1 Table**). We divided proteins into three regions (interface, rim and surface) and analyzed the amino acid composition for each region. Overall, we could clearly distinguish these three regions based on their composition, with polar residues favored at the surface and hydrophobic residues more abundant in rim and interface regions (**Fig 2B**). Differences in each of these regions between BE, EE and BB complexes were more subtle (**Fig 2B and 2C**). While some residues (Trp, Phe and Lys) were enriched in the rim area in BE complexes, only differences in Leu composition were detected in the interface area (**S3 and S4 Figs**). This fact may be related to the higher conservation of the interface compared to the rim or surface regions [28, 29]. It is also important to note that "affinity-defining" positions, located in the interface, are highly optimized whereas "specificity-defining" positions are usually non-optimal and are located at the rim area [16]. Hence, bacteria proteins may preferentially modify the rim area to discriminate between self and non-self interactions.

Despite maintaining the same composition at the interface level, the interaction pattern between amino acids was substantially different between complex types. We evaluated amino-acid interactions and compared the connectivity network of BE complexes with that of EE and BB complexes. In general, we found that the contribution, in terms of the number of interactions, for each amino acid in BE complexes was significantly correlated to EE but not to BB complexes (**Fig 2D**). The correlation between the network of interactions for each amino acid (**Fig 2E and 2F**) and their organization (**S5 Fig**) also confirmed that BE complexes were more similar to EE than BB complexes. Random resampling confirmed that our measurements were not affected by sample size (**Fig 2F**, insert). These results support the theory that bacteria may use molecular mimicry to interact with host proteins. According to the mimicry hypothesis, bacteria can partly or completely imitate the structure of host proteins by mechanisms of gene transfer and/or convergent evolution using a strategy called 'molecular mimicry' [30, 31]. Bacterial proteins competitively bind to the target host site [32, 33] and redirect host hub proteins away from their pathway [34, 35]. This strategy does not necessarily involve changing the entire structure of proteins but only certain residues in the interface or rim areas [34, 35]. These bacterial proteins target host processes involved in cell adherence and invasion, which are essential for infection and explain why certain bacteria display strict host selectivity [36]. However, mimicry has been observed mostly on a case-by-case basis, using sequence or structure similarity [2, 37] or by solving isolated complexes [38].

While the evidence supporting structural mimicry is strong, we noticed clear differences between BE and EE complexes at the amino acid interaction level (**Fig 2E**). For example, Arg interactions had different preferences: Arg-Glu interactions were preferred in BE complexes, whereas Arg-Asp interactions were preferred in EE complexes. This might reflect an evolutionary adaptation, as Glu residues are preferred in eukaryotic interfaces compared to prokaryotic interfaces and vice versa for Asp residues (**S3 Fig**, p = 0.016). In these lines, when analyzing directionality in BE interactions, we observed that some amino acids were frequently targeted at the bacterial interface (Tyr, Arg, Leu and Gly), whereas others were mainly targeted at the eukaryotic interface (Trp, Lys and Met; **Fig 2G**), being Trp was the most conspicuous case (**S6 Fig**). We noticed that Trp-Asp and Trp-Glu interactions were more common in BE

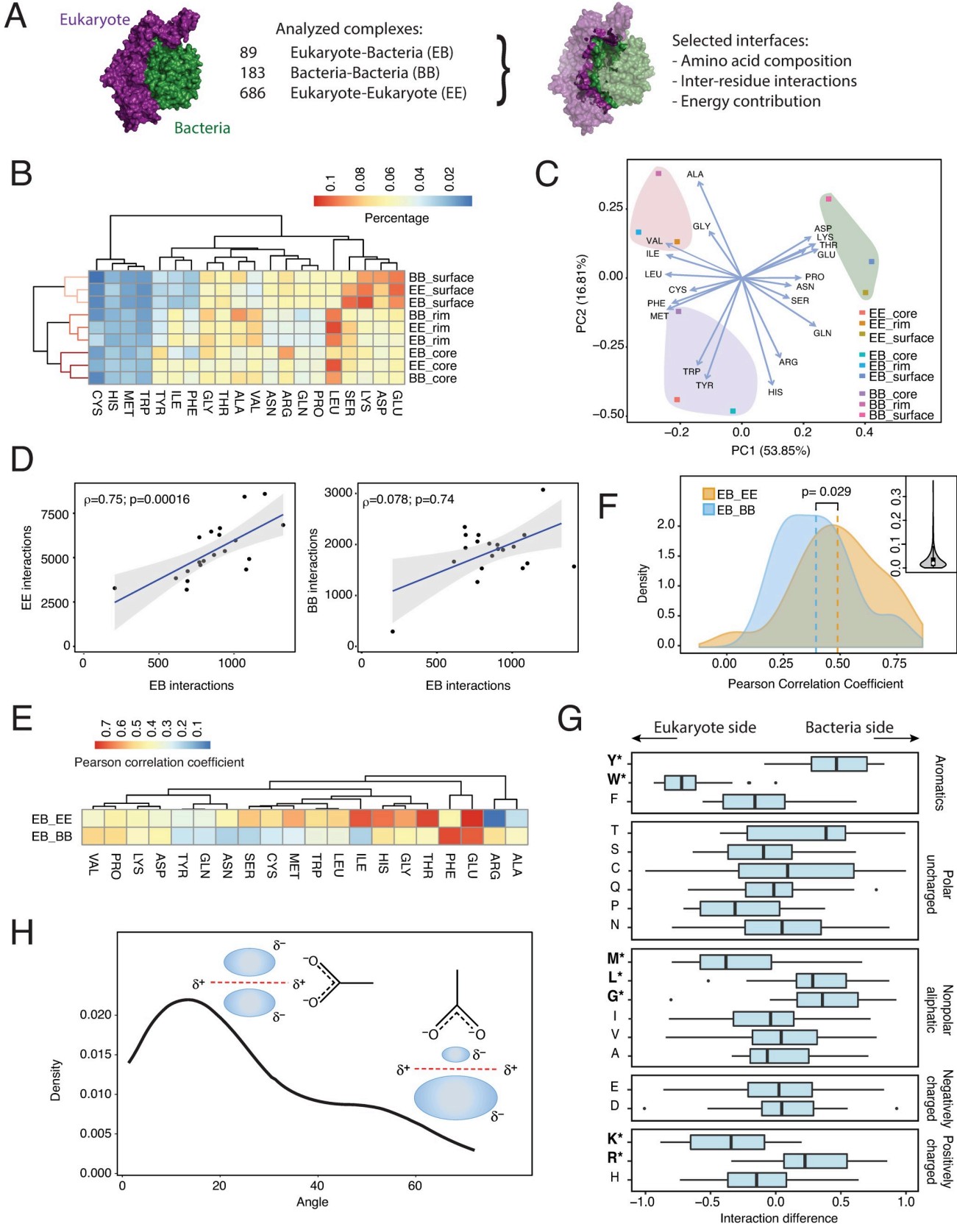

**Fig 2. Analysis of protein complexes.** (**A**), To compare the structural determinants of bacteria-eukaryote (BE) complexes, 89 nonredundant complexes were obtained from the PDB and compared to 183 bacteria-bacteria (BB) and 686 eukaryote-eukaryote (EE) complexes. (**B**) Hierarchical clustering and (**C**), principal component analysis of amino acid composition in BE, BB and EE protein complexes. All percentages were controlled by the amino acid structure of the protein. (**D**) To characterize the interaction pattern in BE complexes, the number of interactions for each amino acid in BE complexes was plotted against EE and BB complexes. Regression lines were calculated using the Spearman rank-correlation approach to control for the impact of extreme values. (**E**) Hierarchical clustering of the interaction pattern for each amino acid. The correlation coefficient for each amino acid was calculated comparing the number of interactions with all other amino acids in BE complexes against EE and BB complexes. (**F**) Distribution of Pearson correlation coefficients as calculated in panel E for BE complexes against EE and BB complexes. The inner panel shows the p-value distribution in a resampling control to correct for effect size (see Materials and Methods) (**G**) Directionality for each amino acid ($D_i$) in BE interactions was calculated as the relative difference in the number of interactions ($N$) in both directions $D_i = \frac{N_i^B - N_i^E}{N_i^B + N_i^E}$. (**H**) Distribution of the angle measured for all anion-pi interactions involving Trp in BE complexes.

complexes (25% of proteins had at least 1 anion-pi interaction) compared to the PDB interactome (less than 10% of proteins had at least 1 interaction) [39]. Asp and Glu were preferred in the bacterial interface, while Trp was mostly located in the eukaryotic interface, which coincides with Trp being more abundant in eukaryotes than bacteria (p-values 0.10 and 0.012, for core and rim, respectively). Furthermore, Trp in the eukaryotic interface had a higher contribution to complex stability compared to Trp in the prokaryotic interface (**S7 Fig**), suggesting that those interactions would contribute to complex stability. In almost all interactions in BE complexes (95%), Trp interacted with anionic residues through anion-pi interactions, which involves the contact of the negative density of Asp and Glu with the positive density at the edge of the aromatic ring (**Fig 2H**). Collectively, these results confirm that bacteria use molecular mimicry to interact with eukaryotic proteins, but also suggest that such mimicry is imperfect. Hence, although the composition of the central interface is similar across all complexes, the differences observed in its geometry can help discriminate between self and non-self interactions. Also, differences in the rim area would allow to fine-tune the binding. In the next section, we explore the use of imperfect mimicry in the context of host-pathogen interactions.

## Imperfect mimicry in the *Y. pestis–H. sapiens* interactome

During the course of infection, pathogens use proteins to rewire a myriad of biochemical processes [40] that are required for efficient propagation [41, 42]. We recently showed that pathogen proteins engaged in a higher number of interactions with the host also have a major impact on pathogen fitness during infection [8]. Hence, the relevance of pathogen proteins in the infection process is proportional to its ability to reorganize the host interactome. Unfortunately, complexes of bacterial proteins with human targets are largely underrepresented in the PDB database.

To further investigate this issue, we used the *Yersinia pestis-Homo sapiens* interactome and analyzed domain-domain associations (in terms of protein superfamilies) in comparison with the isolated *Y. pestis* and *H. sapiens* networks. Such an approach is justified because organisms mainly use the same 'building blocks' for protein interactions, and the function of domain pairs seem to be maintained during evolution [43, 44]. We observed that an important number of associations are shared between the *Y. pestis-H. sapiens* interactome and the *H. sapiens* interactome (19%) compared with the *Y. pestis* interactome (0.72%, p<0.00001; **Fig 3A**). Consistently, the shared subnetwork (intersection) between BE and EE networks is more densely connected compared to the shared subnetwork between BE and BB networks (**Fig 3B and 3C**). Again, this suggests that the BE interactome is more closely related to the EE rather than the BB interactome.

To further validate these results, we filtered the *Y. pestis-H. sapiens* network with fitness data, which measures the relevance of a given bacterial gene in infection. Using this strategy, we created a subset of domain interactions that have a high impact on the fitness of *Y. pestis*

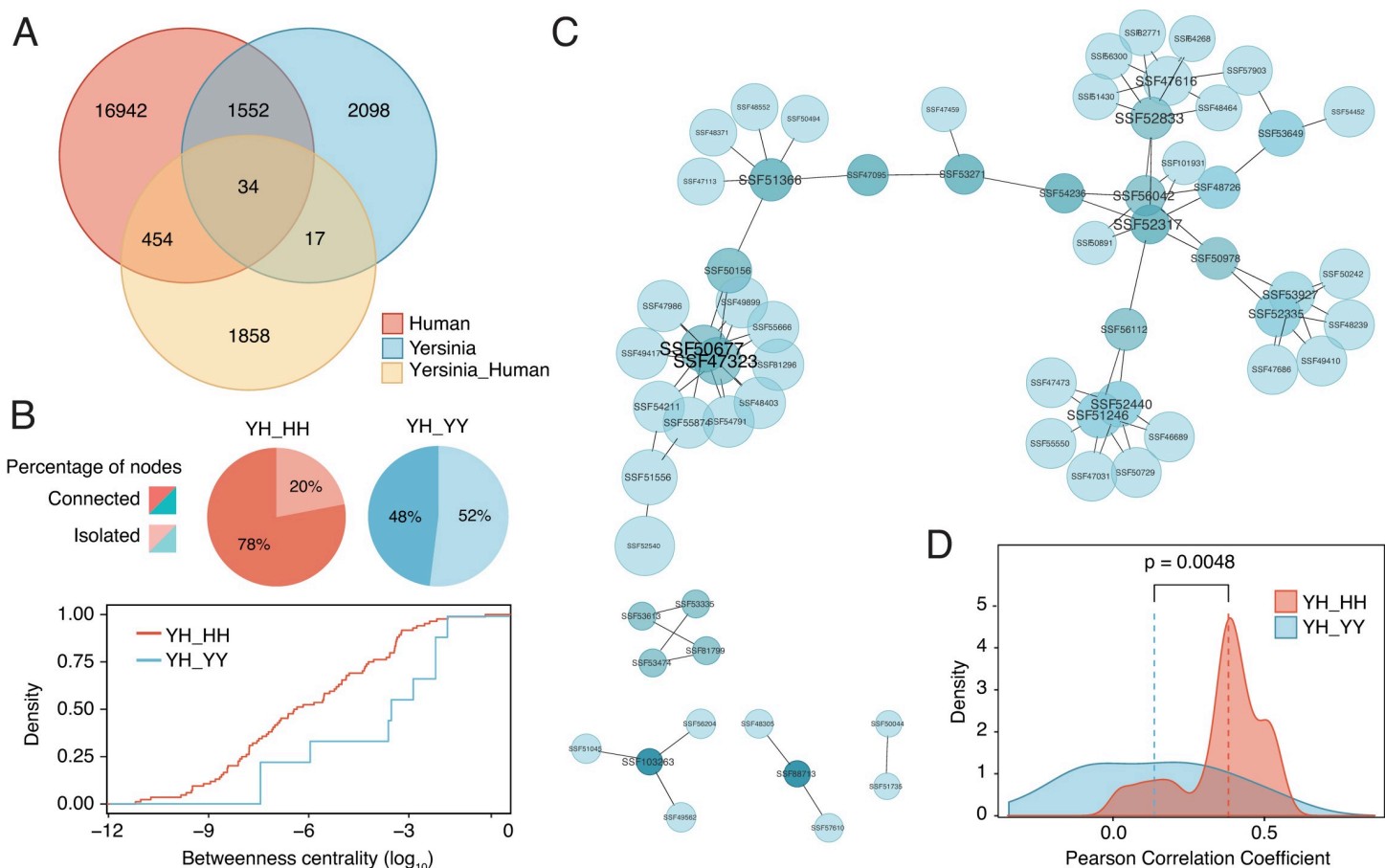

**Fig 3. Structural analysis of protein-protein interactions in the *Yersinia pestis-Homo sapiens* interactome.** (**A**) Venn diagram showing shared and unique domain associations between the *Y. pestis-H. sapiens*, *Y. pestis-Y. pestis* and *H. sapiens-H. sapiens* interactomes. (**B**) Percentage of isolated and connected nodes in the shared subnetworks (intersection) between the *Y. pestis-H. sapiens* interactome and the *Y. pestis-Y. pestis* or *H. sapiens-H. sapiens* interactomes. Cumulative distribution of betweenness centrality in both subnetworks. (**C**) *Y. pestis-H. sapiens* domain association network filtered for *Y. pestis* proteins that have a high contribution to infection fitness (fitness factor < 0.4). Complexes that involve bacterial proteins with a high contribution to infection fitness were modeled and docked to obtain the putative three-dimensional structure. (**D**) Distribution of Pearson correlation coefficients in the filtered network of contacts for all modeled complexes (n = 18). For all amino acids the connectivity matrix and Pearson correlation coefficients were calculated. The plot shows the distribution of correlation coefficients for each amino acid in YH complexes against HH and YY complexes.

during infection. The superfamily associations for such network are significantly enriched in domains related to infection (**Fig 3C, S2 Table**). When possible, we modelled the three-dimensional structure of the proteins involved in this network by sequence similarity and then obtained the structure of the complex by docking simulations (18 complexes). Docking procedures were not highly reliable to delineate interfaces in detail but helped to draw a coarse-grained view of the interactions. To investigate whether the predicted complexes are more similar to BB or EE complexes, the correlation coefficients for the interaction pattern of each residue were obtained (**Fig 3D**). Similar to previous results, we observed that the modelled interactions were more similar to EE complexes than BB complexes (**Fig 2F**). Overall, the correlation coefficients were lower when compared to those of the complexes deposited in the PDB, which can be attributed to the predicted nature of these complexes. Hence, although modelled data must be treated with caution, it reflects a general trend that is consistent with previous observations.

## Discussion

Based on the results presented here, we suggest that bacterial effectors have evolved their interfaces to imperfectly mimic eukaryotic complexes. During this process, effectors would have been subjected to two opposing forces: on the one hand, there would be an evolutionary pressure to increase the number of interactions with the host while, on the other hand, effectors would be forced to minimize the number of interactions with other pathogen proteins (**Fig 4A**). The first condition is necessary to increase the pathogen infectivity and its survival inside the host cells. The second one is less obvious but can be explained in terms of protein stickiness, which is higher in eukaryotes than prokaryotes. By mimicking eukaryotic interfaces, effectors become stickier, potentially increasing the number of spurious interactions with other pathogen proteins. Such poisonous interactions may compromise the cell viability; therefore, pathogens must find a balance between increasing infectivity and limiting toxicity. Using imperfect mimicry of eukaryotic interfaces, effectors are able to discriminate between bacteria-bacteria (self) and bacteria-host (non-self) interactions (**Fig 4B**).

The imperfect mimicry of protein interfaces has direct evolutionary consequences: as pathogen effectors mimic eukaryotic complexes to enhance adaptation, the host, in its turn, evolves its proteome to discriminate its own proteins from the pathogen mimics. This creates an arms race for survival between the host and the pathogen. Such behavior is observed in viral infections, particularly in those caused by poxviruses [45]. In a viral infection, host cells phosphorylate the eukaryotic initiation factor 2A (eIF2a) by protein kinase R (PKR) to inactivate translation. To restore translation, poxviruses evolved a protein called K3L that mimics eIF2a and competes with it for PKR phosphorylation. In its turn, primates also evolve eIF2a to discriminate it from K3L, creating a lasting evolutionary circle.

Effectors regulate pathogen adhesion, survival and proliferation in the host and so, they are frequently found to be essential for infection *in vivo* [46]. However, as mentioned before, these proteins are also isolated within the pathogen interactome, which explains why they are often classified as nonessential for the pathogen growth *in vitro* [8]. Unfortunately, most large-scale screening assays aimed to discover new antimicrobials are developed *in vitro*. Using such approaches, most effectors will never be discovered and the potential drugs that could be developed against them will remain unexplored. Our observations, therefore, confirm the need to redefine our discovery pipelines to properly reflect the host environment.

Last but not least, our results suggest that host-pathogen protein-protein interactions are potential targets for a new generation of antimicrobials. If an interaction is required for the pathogen to infect the host, blocking this interaction would help to stop or delay the infection. Hence, compounds designed to interfere with such host-pathogen interactions would behave as antimicrobials. This idea has been suggested by us previously but also by other authors. The results presented in this paper conceptually takes us one step further in this effort. We propose that bacteria use imperfect mimicry to interact with host proteins. In other words, the network of interactions in host-pathogen PPIs is different from the own host network. Because of this, we could think of compounds, either peptidomimetics or small molecules, that interact with the bacteria interface without disrupting the host complexes. Our observations suggest that such compounds might have therapeutic potential as the interaction with their host targets would be weaker compared with their bacteria counterparts. In summary, treatments interfering with the adhesion and invasion of bacteria to host cells could be used as preventive strategies during surgical procedures or after infection by reducing the resistance of pathogens to known antibiotics by combating their spread in the organism [47].

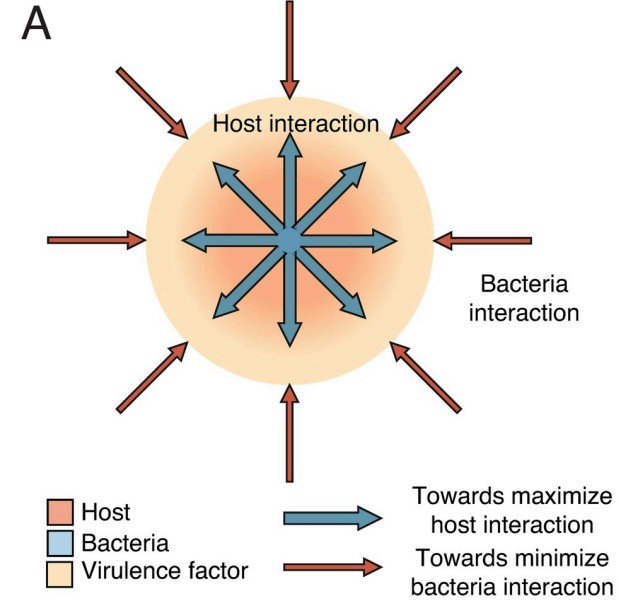

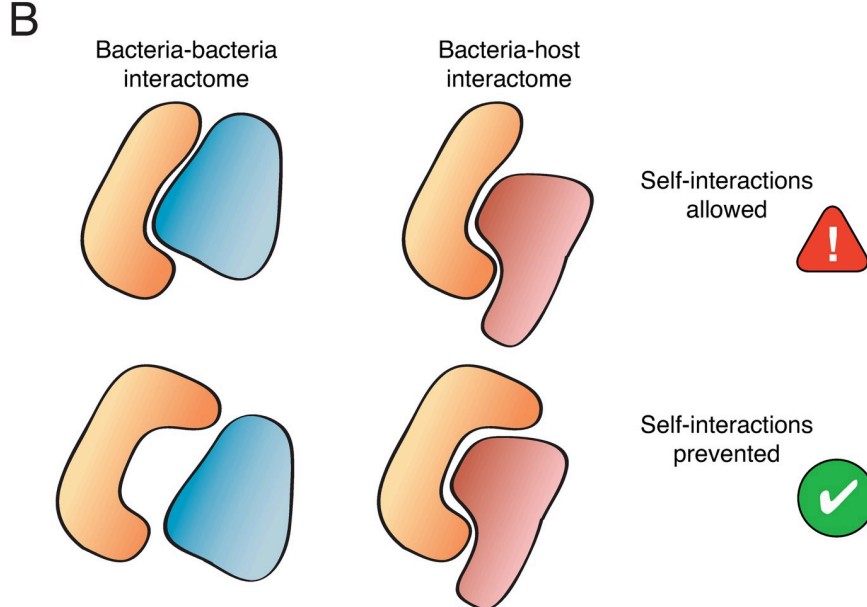

**Fig 4. Organization restraints for host-pathogen interactions.** (**A**) Effectors involved in host-pathogen interactions must optimize the interaction with host proteins while keeping undesired interactions under control within the pathogen interactome. (**B**) This balance is achieved through structural imperfect mimicry. Proteins retain the core interface to strongly interact with the host but modulate its geometry and the rim areas to minimize potentially detrimental self-interactions.

## Materials and methods

### Databases

The *H. sapiens-Y. pestis*, *H. sapiens-B. anthracis* and *H. sapiens–F. tularensis* interactomes were obtained by yeast-two-hybrid (Y2H) and downloaded from IntAct as reported in the study performed by Dyer et al. [48]. In Y2H, cells produce high amounts of proteins that can lead to

spurious interactions (false positives). Also, Y2H can fail to detect some interactions due to limitations of the screening (false negatives). The latter can happen when protein fusion disrupts the interaction interface or interferes with protein folding and when proteins fail to localize to the nucleus. Also, in host-pathogen interactions, both proteins must be present in the same cellular location to interact. This can generate a detection bias: some interactions may not be biologically significant while others can remain undetected. To control for biases in the Y2H, a control dataset was built using the PHISTO database: http://www.phisto.org [49], including the following organisms: *Escherichia coli*, *Pseudomonas aeruginosa*, *Salmonella enterica* and *Shigella flexneri*. The interactions in the control database were defined using different methodological approaches, including pull-down, affinity coimmunoprecipitation and affinity chromatography, among others.

Bacteria-eukaryote (BE, 89 complexes), bacteria-bacteria (BB, 183 complexes) and eukaryote-eukaryote (EE, 686 complexes) were obtained from the Protein Data Bank (PDB). The codes for all proteins selected in the study are included in S1 Table. We acknowledge that the PDB is biased by (i) the preferences of individual investigators and (ii) the physicochemical nature of proteins, which defines the probability of proteins to crystalize. These reasons may explain the higher abundance of EE complexes in the PDB compared to EB and BB complexes. Differences in size between proteins deposited in the PDB and the corresponding genomes have been also reported [50]. Despite these biases, the interface size, measured as the number of interactions per molecule, in all complexes included in our dataset is similar (differences not significant, $p > 0.05$ using a Mann–Whitney U-test).

The *Y. pestis* and *H. sapiens* interactomes were obtained from the String database [51]. Only highly reliable interactions were included (experimentally validated interactions with score > 700). The *Y. pestis* interactome has 3701 interactions for 3973 distinct protein-coding genes and the *H. sapiens* interactome has 18982 interactions for 19566 distinct protein-coding genes. Hence, the number of interactions per protein is similar and we do not expect a bias related to relative size.

EffectiveDB (http://effectors.org/) was used to classify proteins between effectors and non-effectors, using a stringent value of 0.95 [52]. Fitness values were obtained from [53] using transposon sequencing (Tn-seq) and calculated as the ratio of the rates of population expansion for the two genotypes after infection of *Y. pestis* in a mouse model. In total, 1.5 million independent insertion mutants were screened with a coverage of $\sim 70\%$ of the *Y. pestis* genome. Protein superfamilies of *Y. pestis* and *H. sapiens* were obtained from UniProt. Structural parameters were obtained from [54] (alpha helix, beta-sheet and coil propensity), [55] (aggregation propensity), [56] (disorder propensity) and [57] (nucleic acid binding propensity).

## Interface definition and calculation of contact maps

Residue Interaction Networks (RINs) represent amino acid residues as nodes in a network. If two residues interact (based on spatial distance) they are connected by edges between them. Hence, we decided to compare the connectivity profiles of protein complexes and used them as interaction fingerprints. The interface, rim and surface regions were defined using a python script developed by the Oxford Protein Informatics group, which is freely available (http://www.stats.ox.ac.uk/~krawczyk/GetContacts.zip). Briefly, the interface residues were defined as those in close contact between two molecules in a given complex (4.5 Å). Rim residues were not engaged in intermolecular contacts but were close to the interface (contact between molecules < 10 Å) and can have a more subtle effect on binding. Surface residues were determined as residues not present in either the rim or interface region that display a surface

accessible area greater than 20 Å$^2$. We considered that Trp residues were engaged in anion-pi interactions when the distance between the centroid of the aromatic ring and the anion was between 2–5 Å. Anion-pi distances and interaction angles (defined between 0˚ and 90˚) were measured in Pymol. Contact maps were generated in R (version 3.4.4) using the function *cmap* included in the bio3d package [58].

### Modelling and docking of *Y. pestis*-*H. sapiens* complexes

Proteins involved in complexes were retrieved from the PDB when possible. Otherwise, the three-dimensional structure was modelled using Modeller (version 9.21) [59] as long as a template had homology higher than 30%, spanning more than 75% of the protein length. Docking was performed using Frodock (version 2.0) with default parameters [60]. The interface for complexes ranked highest in the docking score was analyzed using the pipeline described before.

### Network analysis

All protein networks were analyzed with Cytoscape (version 3.6.1) [61], and statistical calculations were performed in R (version 3.4.4). The degree ($k$) of a node $i$ is defined as the number of edges linked to $i$. To compare the node degree between two networks, we define the normalized degree as k/(n-1) where $n$ is the number of nodes in the network. Betweenness centrality ($C_b$) was computed as follows:

$$C_b(i) = \sum_{s \neq i \neq t} \frac{\sigma_{st}(i)}{\sigma_{st}}$$

where $s$ and $t$ are nodes in the network different from $i$, $\sigma_{st}$ denotes the number of shortest paths from $s$ to $t$, and $\sigma_{st}(i)$ is the number of shortest paths from $s$ to $t$ that $i$ lies on. The intersection of two networks was calculated using the merge function (intersection) in Cytoscape.

### Statistical analyses

Unless otherwise specified, all p-values were calculated using the Mann–Whitney U-test and considered significant when p<0.05 (see Figure Legends for further details). The $\chi^2$ test was used to determine whether there is a significant difference between the expected and the observed frequencies in two categories. In all cases, two-sided tests were used with a testing level $\alpha$ = 0.05. Random resampling was used to create balanced datasets and determine the effect of sample size. To generate balanced datasets, the majority class was randomly undersampled while the minority class was oversampled. Using this strategy, 500 balanced datasets were generated to assess the effect size.

## Supporting information

**S1 Fig. Strategies to control potentially detrimental interactions in the pathogen interactome.** Pathogens could (A) regulate the expression of protein effectors or (B) compartmentalize the interactions in protein condensates for timely delivery upon infection. As observed in the boxplots in the lower panel, no clear significant differences were observed in all three organisms studied. P-values were calculated using the Mann-Whitney U test for comparing pairs of independent samples.
(TIFF)

**S2 Fig. Fitness impact and differential expression of bacterial proteins belonging to the BE set.** To validate the role of bacterial proteins in the BE dataset during infection, we used blastp against the BacFitBase database (http://www.tartaglialab.com/bacfitbase/) to predict the fitness score associated with each protein. The fitness defect associated to these proteins is larger than a null effect (p = 0.0005, negative values indicate a reduction on pathogen fitness), suggesting that the BE dataset is associated with pathogenesis. Also, we analyzed the mRNA expression data using the DualSeqDB database (http://www.tartaglialab.com/dualseq/). Proteins belonging to the BE dataset are significantly upregulated in infection (p<0.0001) suggesting its implication in bacterial pathogenesis.
(TIFF)

**S3 Fig. Amino acid composition of interface, rim and surface regions in bacteria-eukaryote complexes.** Percentage of each amino acid in the interface, rim and surface regions of bacteria-eukaryote complexes. Each region is subdivided in bacteria (B, red) and eukaryote counterparts (E, blue).
(TIFF)

**S4 Fig. Amino acid preference in interface, rim and surface regions of bacteria-eukaryote complexes.** Venn diagram displaying the differential amino acid composition of interface, rim and surface areas in bacteria-eukaryote complexes. Significant amino acid composition differences between bacteria and eukaryote counterparts in bacteria-eukaryote complexes were calculated using a Mann-Whitney U test and considered significant when p < 0.05.
(TIFF)

**S5 Fig. Comparison of interaction patterns in bacteria-eukaryote complexes compared to bacteria and eukaryote complexes.** Dendrograms for each complex type (bacteria, eukaryote and bacteria-eukaryote) were built according to the interaction pattern of each amino acid using the Ward's minimum variance method. For each dendrogram, four groups (red, green, orange and blue) were defined using k-means clustering. Then, tanglegrams were built to compare the congruence between dendrograms (bacteria-eukaryote compared to bacteria and eukaryote). In the figure, congruence is depicted by the number of colored lines mapping common elements between same groups in different dendrograms. The quality of the alignment of the two dendrograms (entanglement) is also reported as a quantitative measure of congruence. Entanglement is measured between 1 (full entanglement, high congruence) to 0 (no entanglement, null congruence). Tanglegrams were built using the *dendextend* package in R.
(TIFF)

**S6 Fig. Amino acid connectivity analysis of bacteria-eukaryote, bacteria and eukaryote complexes.** Number of interactions for each amino acid (measured as the percentage relative to the total number of interactions) in bacteria-eukaryote (BE, red), bacteria (BB, blue) and eukaryote (EE, green) complexes. Each pair of letters correspond to an amino acid in the interaction pair. The scale measures the percentage of a given interaction over all detected interactions.
(TIFF)

**S7 Fig. Residue contribution to the complex stability by alanine scanning in bacteria-eukaryote complexes.** Each residue in the interface of bacteria-eukaryote complexes was mutated to alanine and the change in complex stability was calculated using FoldX. The impact in complex stability for mutations in bacteria (red) and eukaryote (blue) interaction counterparts is compared.
(TIFF)

**S1 Table. Uniprot codes of protein structures included in this study.**
(XLSX)

**S2 Table. List of superfamily associations significantly enriched in domains related to infection.**
(XLSX)

**S3 Table. List of cluster associations related to Fig 1C.**
(XLSX)

## Author Contributions

**Conceptualization:** Natalia Sanchez de Groot, Marc Torrent Burgas.

**Formal analysis:** Natalia Sanchez de Groot.

**Funding acquisition:** Marc Torrent Burgas.

**Methodology:** Natalia Sanchez de Groot, Marc Torrent Burgas.

**Supervision:** Marc Torrent Burgas.

**Writing – original draft:** Natalia Sanchez de Groot, Marc Torrent Burgas.

**Writing – review & editing:** Natalia Sanchez de Groot, Marc Torrent Burgas.

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
