## [Decision Letter · Decision Letter 0]

19 Aug 2020

Dear Dr. Torrent,

Thank you very much for submitting your manuscript "BACTERIA USE STRUCTURAL IMPERFECT MIMICRY TO HIJACK THE HOST INTERACTOME" for consideration at PLOS Computational Biology.

As with all papers reviewed by the journal, your manuscript was reviewed by members of the editorial board and by several independent reviewers. In light of the reviews (below this email), we would like to invite the resubmission of a significantly-revised version that takes into account the reviewers' comments.

We cannot make any decision about publication until we have seen the revised manuscript and your response to the reviewers' comments. Your revised manuscript is also likely to be sent to reviewers for further evaluation.

Sincerely,

Arne Elofsson

Deputy Editor

PLOS Computational Biology

Arne Elofsson

Deputy Editor

PLOS Computational Biology

Reviewer's Responses to Questions

**Comments to the Authors:**

Reviewer #1: The manuscript “Bacteria use Structural imperfect mimicry to hijack the host interactome” by Natalia de Groot and Marc Burgas focuses on understanding how pathogen proteins bind their host counterparts, which is important to gain insight on how bacteria can infect, survive and proliferate inside cells. Overall, the proposed ideas are interesting, and can be of interest to the readers. But I still have some reservations for this manuscript, which I would like the authors to consider, if a revision is permitted.

Major points:

1. I am not sure I fully appreciate the idea of mimicry. As I understand, mimicry means pathogens mimic the host protein, but the paper looks at it in terms of individual amino acids, and to say that positive and negative charges interact doesn't seem like enough to conclude the pathogen is trying to mimic the host (short sequences or confirming host interaction sites are the same would be better).

2. Though the interactome of BE is validated, which is more similar to EE than BB, authors need to show any non-pathogenic BE won’t have the same similarity to EE of pathogenic BE.

3. If the interactome database for non-pathogenic BE is scarce then the authors should not discuss the pathogenicity as a mimicry of EE. This undermines the importance of the pathogenicity validation besides the analytical resemblance of BE to EE.

4. Authors claim that BE interactome includes pathogenic effectors, which needs to be substantiated in detail on the related mechanism from the dataset, or verify from the existing antimicrobial effect on disturbing the interfaces of BE.

5. With no pathogenicity, the resemblance of BE to EE might be a common phenomena to share the evolutionary residue when overwhelming number of proteins being interacting.

6. I appreciate the innovative idea put forward in the data mining process and analysis. But still the strategies needs to be detailed with respect to the development of antimicrobial agents from BE, if the claim is true.

Overall, I think the manuscript is not fully developed with lots of lingering questions still. I feel authors claim on the “mimicry of EE by pathogen BE” sounds nebulous as there is no proper comparison performed with non-pathogenic BE due to lack of data. As well, there is no elaboration on how antimicrobial development would be utilized by data mining, including for very specific rim, core and individual AA interaction in BE. Without these details and additional analyses, I think the findings from this study may be overstated.

**Have all data underlying the figures and results presented in the manuscript been provided?**

Reviewer #1: Yes

PLOS authors have the option to publish the peer review history of their article (what does this mean?). If published, this will include your full peer review and any attached files.

Reviewer #1: No
---

## [Decision Letter · Decision Letter 1]

23 Sep 2020

Dear Dr. Torrent,

We are pleased to inform you that your manuscript 'BACTERIA USE STRUCTURAL IMPERFECT MIMICRY TO HIJACK THE HOST INTERACTOME' has been provisionally accepted for publication in PLOS Computational Biology.

Best regards,

Arne Elofsson

Deputy Editor

PLOS Computational Biology

Arne Elofsson

Deputy Editor

PLOS Computational Biology

Reviewer's Responses to Questions

**Comments to the Authors: **

Reviewer #1: Authors have thoroughly addressed my concerns, and I recommend this manuscript for publication, which should be well received by the broad readership of PLOS Computational Biology.

**Have all data underlying the figures and results presented in the manuscript been provided?**

Reviewer #1: Yes

PLOS authors have the option to publish the peer review history of their article (what does this mean?). If published, this will include your full peer review and any attached files.

Reviewer #1: No

---

## [Editor Report · Acceptance letter]

16 Nov 2020

PCOMPBIOL-D-20-01176R1 

BACTERIA USE STRUCTURAL IMPERFECT MIMICRY TO HIJACK THE HOST INTERACTOME

Dear Dr Torrent,

I am pleased to inform you that your manuscript has been formally accepted for publication in PLOS Computational Biology. Your manuscript is now with our production department and you will be notified of the publication date in due course.

With kind regards,

Nicola Davies
